# Identification of O-glycosylation related genes and subtypes in ulcerative colitis based on machine learning

**Yue Lu[☉], Yi Su[☉], Nan Wang[☉], Dongyue Li, Huichao Zhang, Hongyu Xu[ID]***

Department of Gastroenterology, The First Affiliated Hospital of Harbin Medical University, Harbin, Heilongjiang, China

[☉] These authors contributed equally to this work.

* xuhongyu@ldy.edu.rs

**Data Availability Statement:** The data underlying the results presented in the study are available from the GEO database (www.ncbi.nlm.nih.gov/geo/) at the following accession numbers: Accession Number GSE75214 - https://www.ncbi.

## Abstract

Ulcerative colitis (UC) is an immune-related inflammatory bowel disease, with its underlying mechanisms being a central area of clinical research. O-GlcNAcylation plays a critical role in regulating immunity progression and the occurrence of inflammatory diseases and tumors. Yet, the mechanism of O-GlcNAc-associated colitis remains to be elucidated. To this end, the transcriptional and clinical data of GSE75214 and GSE92415 from the GEO database was hereby examined, and genes MUC1, ADAMTS1, GXYLT2, and SEMA5A were found to be significantly related to O-GlcNAcylation using machine learning methods. Based on the four hub genes, two UC subtypes were built. Notably, subtype B might be prone to developing colitis-associated colorectal cancer (CAC). This study delved into the role of intestinal glycosylation changes, especially the O-GlcNAcylation, and forged a foundation for further research on the occurrence and development of UC. Overall, understanding the role of O-GlcNAcylation in UC could have significant implications for diagnosis and treatment, offering valuable insights into the disease's progression.

## Introduction

Ulcerative colitis (UC), an inflammatory bowel disease, has been a persistent challenge for patients over decades. Elucidating the deeper and more precise mechanisms behind UC has been a key focus in clinical research. Immune response holds considerable significance in the occurrence and development of UC [1]. The pathogenesis of UC includes various components of immunoinflammatory pathways related to the intestine, including antigen recognition, immune response, epithelial barrier, and intestinal microbiota [1–3]. In addition, various types of immune cells, such as antigen-presenting cells (dendritic cells and macrophages), T helper cells, regulatory T cells, and natural killer T cells, play vital roles in the pathogenesis of UC by regulating, inhibiting, and maintaining inflammation [4–6]. Addressing immune-related issues is now a critical component of the fundamental research regarding ulcerative colitis. Previous studies have demonstrated a strong link between glycosylation and both colon inflammation and the colonic immune response [7–9].

nlm.nih.gov/geo/query/acc.cgi?acc=GSE75214
Accession Number GSE92415 - https://www.ncbi.nlm.nih.gov/geo/query/acc.cgi?acc=GSE92415

**Funding:** This study was funded by the Natural Science Foundation of Heilongjiang Province, LH2020H037, Hongyu Xu.

**Competing interests:** The authors have declared that no competing interests exist.

Glycosylation is a reversible post-translational modification that involves the enzymatic covalent attachment of monosaccharides or glycans to proteins. This process is known as glycosylation [10]. As an essential modification of proteins, protein glycosylation mainly includes N-glycans, O-glycans, and other type [10, 11]. O-GlcNAcylation, also known as O-glycosylation, plays an essential role in regulating innate immune cell function, cell metabolism, and the occurrence of inflammatory diseases and tumors [12]. Research has revealed that the levels of O-GlcNAcylation on proteins alter when innate immune cells are stimulated during inflammatory states [13–16]. Consequently, the disruption of O-GlcNAcylation balance in the body can lead to a range of diseases, encompassing intestinal inflammatory disorders, diabetes, neurodegeneration, and even tumors [17–19]. The relationship between intestinal O-GlcNAcylation and ulcerative colitis has attracted increasing attention [9, 12, 20–22].

Intestinal mucosal injury is the most direct manifestation of ulcerative colitis. Most intestinal glycans are mucin-type O-glycans, making up 80% of the mass of human MUC2, the most prevalent intestinal mucin [23]. Intestinal epithelial O-glycans can directly regulate microorganism interactions by providing ligands for bacterial adhesins and nutrients for bacterial metabolism [22]. Various evidence has supported the strong connection between O-glycosylation and ulcerative colitis. Identifying glycosylation biomarkers and their expression changes is essential for diagnosing ulcerative colitis, predicting its progression, and assessing potential complications.

In this study, bioinformatics methods were employed to explore the role of O-GlcNAcylation in developing ulcerative colitis. Furthermore, multiple machine learning methods were used to classify UC into two subtypes based on key genes, guiding the choice of UC treatment and prognostic judgment of UC.

## Materials and methods

### Datasets and sample selection

The following criteria from the Gene Expression Omnibus (GEO) database (www.ncbi.nlm.nih.gov/geo/) retrieval of UC microarray datawere included: a) data from the same sequencing platform to generate expression of two different spectra; b) inclusion of human test samples only; and c) a minimum of ten samples per groups. Finally, two datasets, namely GSE75214 (provided by the GPL6244 platform) and GSE92415 (provided by the GPL13158 platform), were hereby incorporated. The GSE75214 database contains intestinal mucosal biopsies obtained endoscopically from UC patients (n = 97) and healthy controls (n = 11), followed by microarray analysis to assess gene expression. GSE92415 database enrolled 21 healthy subjects and 162 UC patients, including baseline before treatment (n = 87) and post-treatment individuals (n = 75), to evaluate the effect of golimumab (GLM) during induction treatment in moderately to severe UC. In this study, only 87 untreated UC samples and 21 healthy control samples were selected.

### Merge and deduplication of datasets

The genes from both the UC patients and healthy individuals in the GSE75214 dataset were merged with those from GSE92415 to form a comprehensive data set. The batch effect was then eliminated to minimize discrepancies between the different datasets, using the R packages "limma" and "sva" [24, 25]. There were a total of 216 samples and 16467 genes after combination (S1 File). S1A and S1B Fig present the data before and after the merger, respectively (S1A and S1B Fig). To eliminate the adverse effects caused by singular sample data, the datasets was homogenized by R packages "preprocessCore". S1C and S1D Fig illustrate the data before and after normalization, respectively (S1C and S1D Fig).

### Identification of differential expressed genes

The genes that met the criteria of the adjusted P-value < 0.05 and |log fold change (FC)| > 1.0 or 0.5 were considered DEGs using the "limma" package. The volcano plot and the heatmap visualized the DEGs using the "ggplot2" package and "pheatmap" package, respectively.

### Biological function and pathway enrichment analyses

Using the R language "clusterprofiler" package, Gene ontology (GO) and Kyoto Encyclopedia of Genes and Genomes (KEGG) enrichment analyses were performed to identify the potential functions of differential genes and signaling pathways associated with DEGs. GO assays included biological process (BP), Cell component (CC), and molecular function (MF) categories.

### Identification and functional enrichment analysis of O-GlcNAcylation differential genes

The O-GlcNAcylation gene set was downloaded from the MsigDB database (https://www.gsea-msigdb.org/). Subsequently, the gene set between the UC and healthy control groups was extracted and interacted with the O-Glcnacylation gene set to search for differential genes. For this analysis, the R "clusterprofiler" package was utilize [26]. The GO and KEGG enrichment analyses were carried out to derive visual representations of the enrichment results.

### PPI

STRING database (https://string-db.org/) was used to construct Protein-Protein Interaction Networks (PPI) encoded by 7 DEGs to represent the relationships among the 7 differential gene [27].

### Machine learning

LASSO was performed to enhance the predictive accuracy and comprehensibility of the statistical models by employing a regression method for variable selection. Random Forest (RF) is a versatile computational method capable of predicting continuous variables. It is adaptable to various conditions and is known for its high accuracy and sensitivity [28]. Support vector machine (SVM) is a supervised machine learning (ML) method capable of learning from data and making decisions [29].

In this study, three machine learning methods, namely LASSO regression (R-packaged glmnet), Random Forest (R language randomForest), and SVM support vector machine (R-packaged kernel), were used to screen essential differential genes from the seven candidate genes.

### Core genes predicting the disease onset

The receiver operating characteristic (ROC) curve was drawn using the pROC software package (R Package pROC) to evaluate the sensitivity and specificity of four core genes in predicting disease occurrence, with the X-axis indicating "specificity" and the Y-axis representing "sensitivity" [30, 31]. Other gene predictions could obtain different ROC curves. Different areas under the ROC curve (AUC) were obtained, reflecting the gene's strength in predicting disease occurrence.

## Immune landscape of dataset

CIBERSORT (http://cibersort.stanford.edu/) was used to determine the GSE75214 and GSE92415 states of the immune cells infiltrating. Following that, Spearman's method was employed to assess the correlation between the expression of the four pivotal genes and that of immune cells within the dataset samples.

## GSEA analysis of single genes

To characterize the potential functions of the four hub genes, the R clusterProfiler package was used to display the top 20 results of four single-gene GSEA analyses of Reactome [32]. The listed values denoted enrichment scores, with scores above zero indicating a positive correlation between the gene and the pathway, and scores below zero suggesting a negative correlation. The results were then ranked in descending order according to the absolute value of the normalized enrichment score (NES).

## Unsupervised clustering of genes

Based on the four core genes, the R package "ConsensusClusterPlus" was used for unsupervised consensus cluster analysis, identifying two subtypes as optimal, this analysis further highlighted the differential expression of core genes among different types. A p-value less than 0.05 was considered significant. R-package pheatmap was used to draw a heatmap to show the expression differences of the four gene expressions among different subtypes.

## GSVA analysis of different types of pathways

KEGG path and Reactome path were downloaded from the Msigdb database, respectively. The R package GSVA was used to score the paths and to compare the differences between the paths of the two subtypes [33]. Subsequently, an R package, "pheatmap," was adopted for drawing a heatmap to compare the two groups.

## Biological function and pathway enrichment analysis of two subtypes

PCA diagram was used to show the distribution of different subtypes of UC samples, indicating the relationship between two distinct subtypes of UC. Further differential analysis was performed for subtypes, and the selected differential genes were enriched by GO and KEGG. Clusterprofiler was utilized to obtain visual enrichment analysis results.

## Prediction of miRNAs and transcription factors upstream of genes

The regnetwork database (https://regnetworkweb.org/) was used to predict miRNAs and transcription factors (TFs) upstream of genes, with red indicating the core gene. Finally, Cytoscape software was used to construct the network [34].

The R language codes related to bioinformatics methods involved in this study have been uploaded as supplementary information (S2 File).

# Results

## Identification and functional enrichment analysis of DEGs

Upon the merging of the two datasets, namely GSE75214 (provided by the GPL6244 platform) and GSE92415 (provided by the GPL13158 platform), 184 UC samples and 32 healthy controls were ultimately obtained (Table 1).

**Table 1. Sample numbers.**

|  | Healthy Control | Ulcerative Colitis |
|---|---|---|
| GSE75214 | 11 | 97 |
| GSE92415 | 21 | 87 |
| Total | 32 | 184 |

The GSE75214 database contained UC patients (n = 97) and healthy controls (n = 11), GSE92415 database enrolled 21 healthy subjects and 162 UC patients, including baseline before treatment (n = 87) and post-treatment individuals (n = 75), which were excluded from the present research.

The two databases from the GEO database underwent normalization and subsequent merging (S1 Fig). Subsequently, the limma package of R language was used for differential analysis between UC and control, and the differentially expressed genes were screened according to the criteria of |logFC|>1 and adj.P.Val <0.05. The results showed that 449 genes were co-up-regulated and 233 were co-down-regulated (Fig 1A). Heatmap analysis showed significant gene expression differences between UC and the healthy control group. For example, REG and MMP families had significantly high expression in UC, while low expression was in healthy control groups (Fig 1B).

GO enrichment analysis involving BP, CC, and MF showed that differential genes of UC and healthy controls were mainly enriched in leukocyte migration, neutrophil, granulocyte chemotaxis, and regulation of immune processes (Fig 1C–1E). Regarding the KEGG pathway, significant enrichment pathways were TNF, IL-17 signaling pathway, rheumatoid arthritis, NF-κb, and B-cell receptor signaling pathway (Fig 1F). Additionally, UC and healthy controls of DEGs were primarily observed in immune and inflammatory pathways.

## Screening and functional enrichment of O-GlcNAcylation-associated differential genes

The upregulated UC-associated differential genes and O-GlcNAcylation gene sets were extracted and overlapped. Upon overlapping analysis, four common differential genes were identified, including ADAMTS1, MUC1, ST3GAL1, and THBS2 (Fig 2A). The downregulated differential genes and O-GlcNAcylation gene sets between the UC group and the healthy control group were extracted for intersection analysis, and three common differential genes were identified, including SEMA5A, GXYLT2, and GALNT12, after overlapping analysis (Fig 2B).

GO enrichment analysis uncovered that differential genes were mainly enriched in protein glycosylation, biosynthesis, and metabolism of glycoproteins (Fig 2C). The KEGG pathway significantly increased in other glycosylation biosynthetic pathways, Mucin-type O-glycan biosynthesis, and PI3K-Akt signaling pathway (Fig 2D). Furthermore, it was found that the seven differential genes were mainly involved in the glycosylation biosynthesis and metabolism pathways. Moreover, three critical genes, including GXYLT2, GALNT12, and ST3GAL1, were selected and enriched in the KEGG top 5 pathways (Table 2), and their corresponding relationships were labeled (Fig 2E).

## Expression and correlation of the hub DEGs

The expression levels of the seven differentially expressed genes identified through intersection analysis between the UC and healthy control group were visualized using a volcano plot and a heatmap. These visualizations provided a comparative analysis of the gene expression patterns in both groups (Fig 3A). As shown in the figure, the expression of ADAMTS1, MUC1,

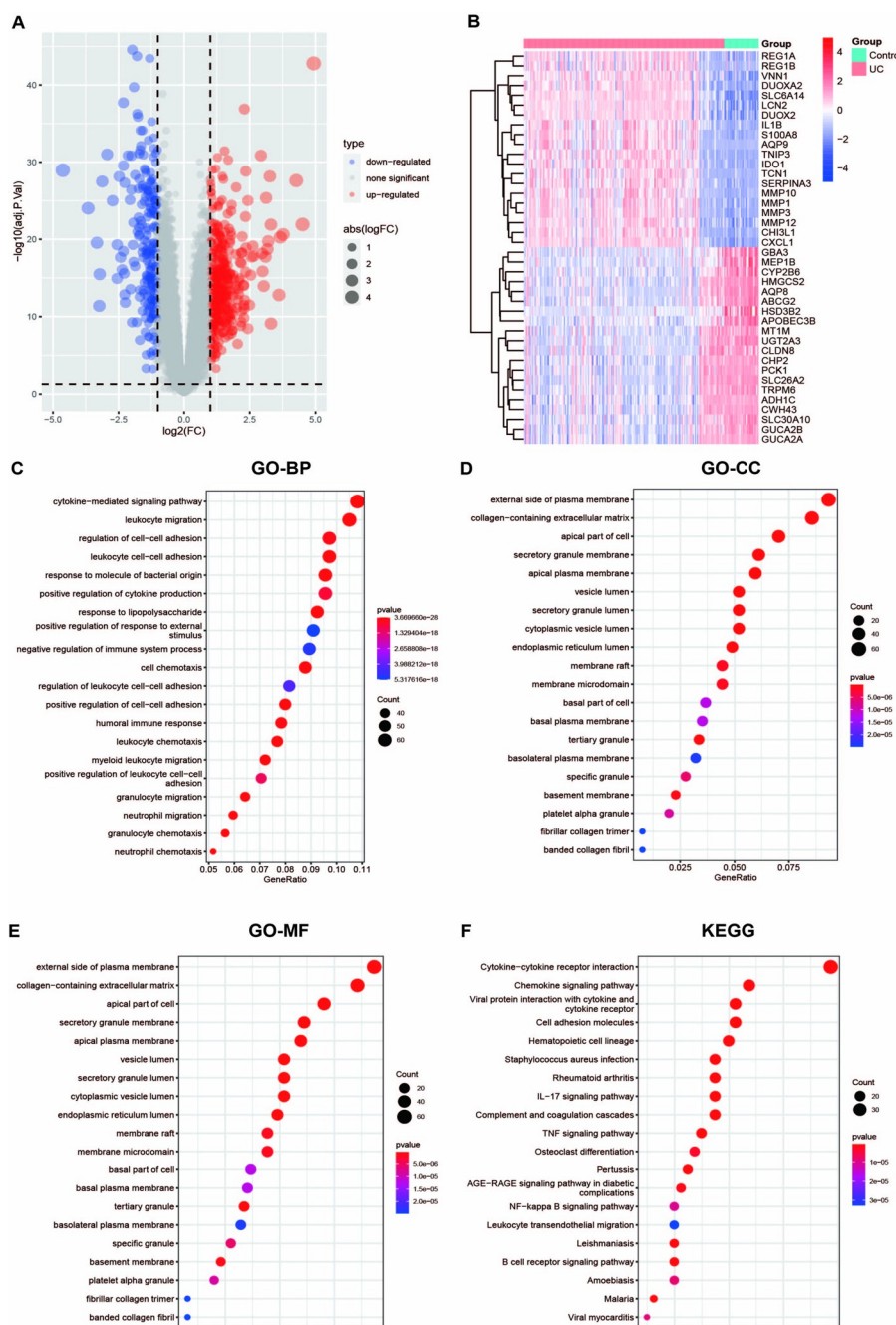

**Fig 1. Identification and functional enrichment analysis of DEGs.** (A) The volcano map showed DEGs from two GEO datasets, UC and health control. (B) The heatmap showed the different genes between UC and healthy controls. The screening criteria were set to |LogFC| > 1 and adj.P.Val < 0.05. (C-E) The enrichment analysis results of GO, including BP, CC, and MF, revealed the underlying functions of DEGs. (F) KEGG revealed the first twenty pathways of differential gene enrichment.

ST3GAL1, and THBS2 were significantly upregulated in UC, while those of SEMA5A, GXYLT2, and GALNT12 were considerably downregulated in UC (Fig 3B). Using the STRING database (https://string-db.org/), PPI Networks encoded by 7 DEGs were constructed. PPI, an interaction network comprising 7 nodes and 18 edges, was visualized using Cytoscape software (Fig 3C).

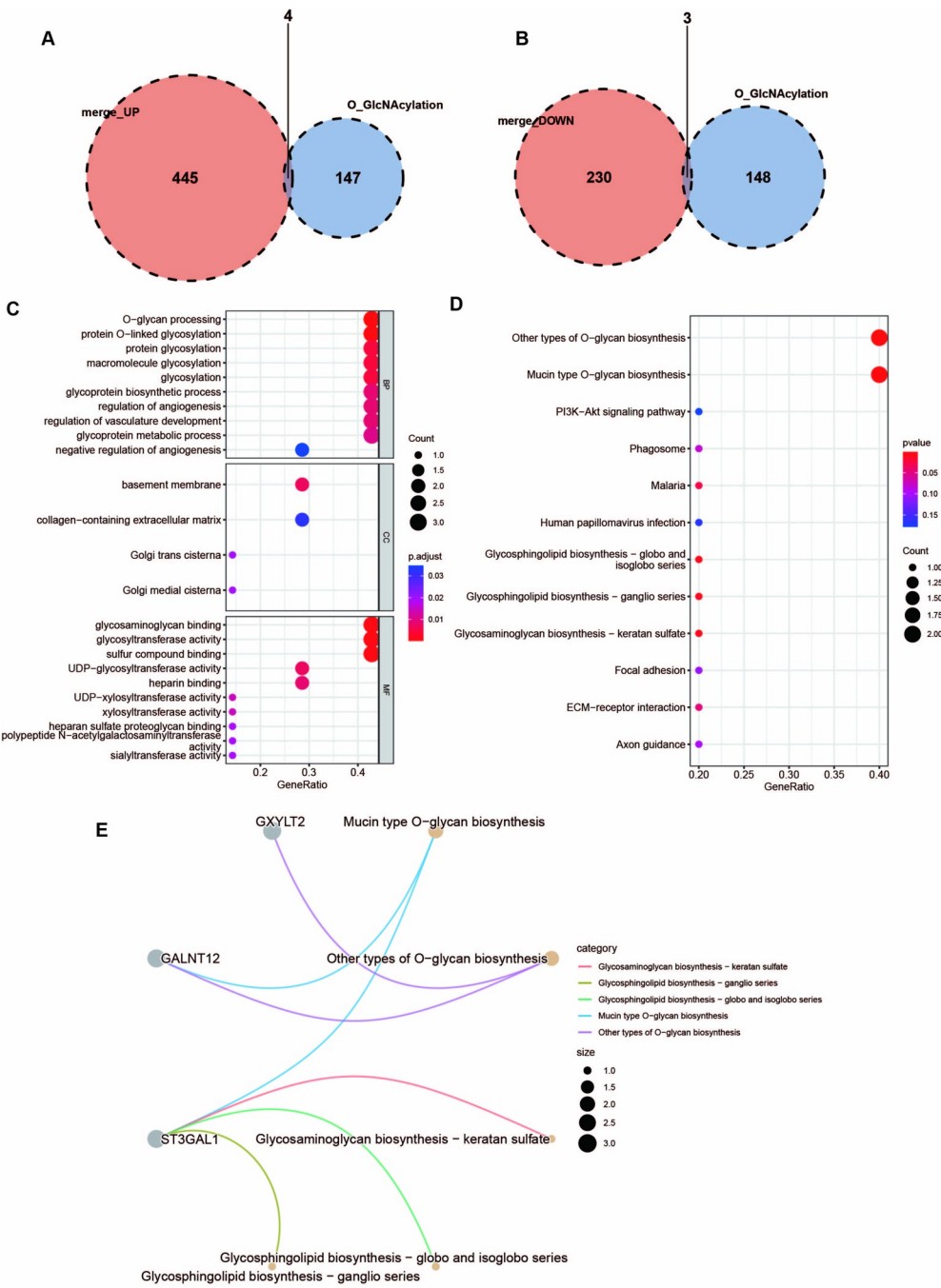

**Fig 2. Screening and functional enrichment of O-GlcNAcylation-associated differential genes.** (**A**) Venn diagram of upregulated differential genes and O- GlcNAcylation gene sets. (**B**) Venn diagram of downregulating differential gene and O-GlcNAcylation gene set. (**C**) The enrichment analysis results of GO, including BP, CC, and MF. (**D**) The KEGG enrichment of DEGs. (**E**) Mapping between the top 5 pathway of KEGG and three differential genes, with different colored lines corresponding to different KEGG pathways.

## Machine learning screening for key differential genes

Herein, the most important features were selected based on three machine learning algorithms to screen out the hub genes with the most guiding value further from the 7 DEGs. The LASSO

**Table 2. The KEGG enrichment analysis table of DEGs.**

| ID | Description | GeneRatio | BgRatio | pvalue | p.adjust | geneID |
|---|---|---|---|---|---|---|
| hsa00512 | Mucin type O-glycan biosynthesis | 2/5 | 36/9180 | 0.000148427 | 0.00152442 | ST3GAL1/GALNT12 |
| hsa00514 | Other types of O-glycan biosynthesis | 2/5 | 47/9180 | 0.00025407 | 0.00152442 | GXYLT2/GALNT12 |
| hsa00533 | Glycosaminoglycan biosynthesis - keratan sulfate | 1/5 | 14/9180 | 0.007603702 | 0.019548115 | ST3GAL1 |
| hsa00603 | Glycosphingolipid biosynthesis - globo and isoglobo series | 1/5 | 15/9180 | 0.008145048 | 0.019548115 | ST3GAL1 |
| hsa00604 | Glycosphingolipid biosynthesis - ganglio series | 1/5 | 15/9180 | 0.008145048 | 0.019548115 | ST3GAL1 |

Table 2 demonstrated that KEGG top 5 pathways ranked in ascending order of p-value, and the relevant information including ID, Description, GeneRatio, BgRatio, pvalue, p.adjust, qvalue, geneID, and Count.

logistic regression algorithm, RF analysis, and SVM algorithm were carried out successively. 4 key genes were selected based on the results of the three algorithms.

Furthermore, LASSO analysis was conducted to identify 5 tag genes, namely ADAMTS1, MUC1, ST3GAL1, SEMA5A, and GXYLT2 (Fig 4A). In the RF analysis, 5 tag genes were selected in order of relative importance, namely SEMA5A, ADAMTS1, MUC1, GXYLT2, and THBS2 (Fig 4B). Moreover 6 tag genes, namely ADAMTS1, SEMA5A, MUC1, GXYLT2, THBS2 and GALNT12, were identified using SVM (Fig 4C). 4 core genes were finally recognized through the interaction of these three algorithms, including GXYLT2, MUCI, ADAMTS1, and SEMA5A (Fig 4D). Subsequently, the correlations among the 4 core genes screened by machine learning were evaluated, with red representing positive and green indicating negative correlations (Fig 4E).

A significant correlation between SEMA5A, GXYLT2, and MUC1 expression was observed, with SEMA5A showing a strong positive correlation with GXYLT2 expression and a strong negative correlation with MUC1 expression. ADAMTS1 was negatively correlated with the expression of GXYLT2 but exhibited no significant correlation with the expression of MUC1. In addition, there was a significant negative correlation between MUC1 and GXYLT2 expression. As indicated by the ROC curves, the four hub genes demonstrated a strong predictive power for UC (GXYLT2 AUC = 0.923, MUC1 AUC = 0.898, ADAMTS1 AUC = 0.955, and SEMA5A AUC = 0.944) (Fig 4F).

## Evaluation of the degree of immune cell infiltration

In this study, the relationship between immune cells in UC was also investigated and the results demonstrated a positive correlation in the expression of Activated B cells, Activated CD4, CD8 T cells, Natural Killer cells, and other immune cells. The expressions of Type 17 helper cell, activated B cell, and activated CD8 T cell were negatively correlated, respectively, while most other immune cells were positively correlated with each other (Fig 5A). CiberSort was employed to futher demonstrate the difference in immune cell infiltration between UC and healthy control group. The results showed that the levels of Activated B cells, Activated CD4, CD8 T cells, Natural Killer cells, and other immune cells in UC patients were significantly higher than those in the standard control group. Besides, no significant difference in the expression of Type 17 helper cell and CD56dim natural killer cells was identified between the UC and healthy control group (Fig 5B).

Furthermore, the connection between 4 core genes and immune cells was also delved into. The findings revealed a significant correlation between the expression of these core genes and activated CD4 T cells, natural killer cells, Type 17 helper cells, and CD56dim natural killer cells. Among them, ADAMTS1 was observed to be significantly positively correlated with the expression of natural killer T cells and activated CD4 T cells while being significantly

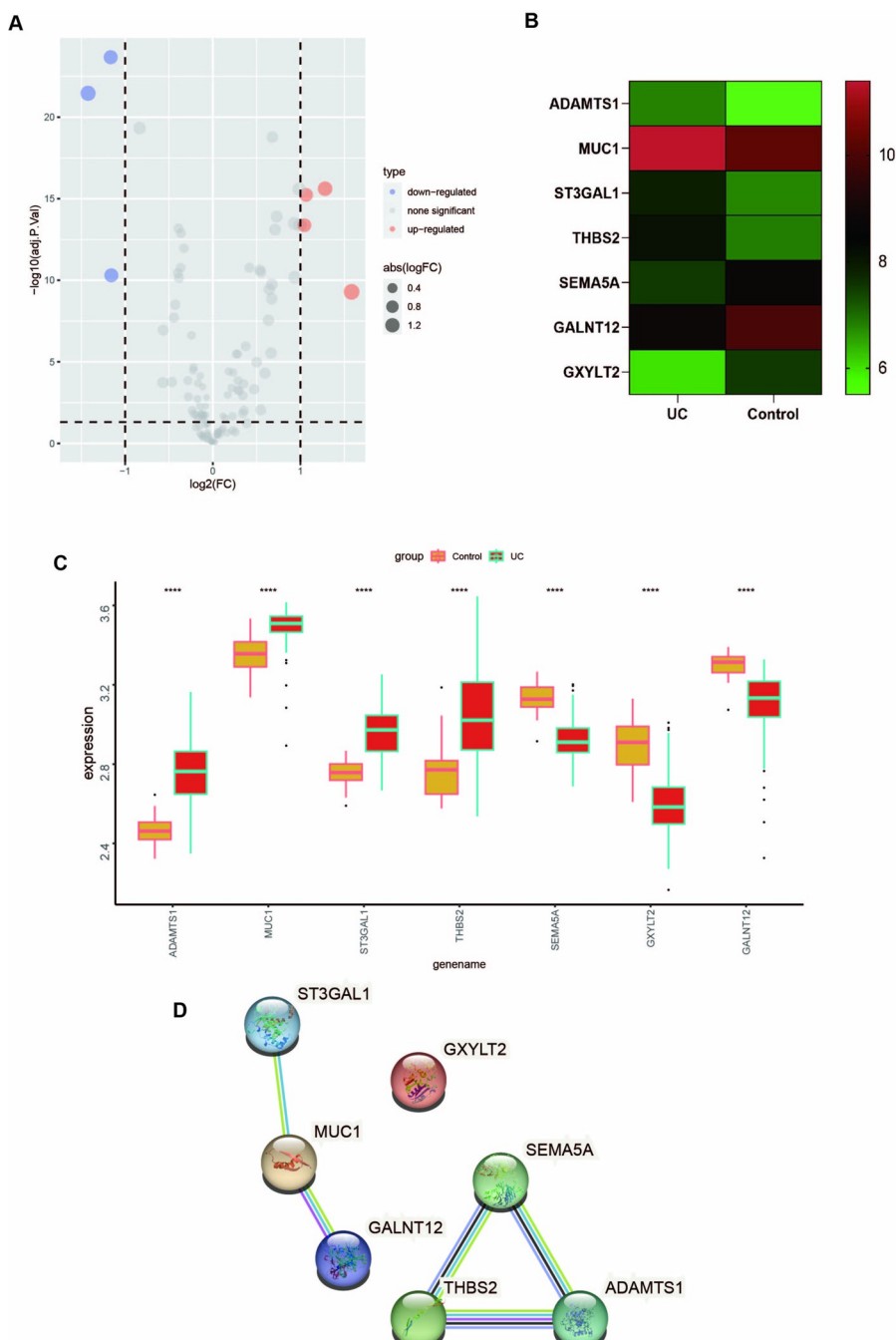

**Fig 3. Expression and correlation of the hub DEGs. (A)** The volcano map of the seven differential genes presented separately. **(B)** Expression analysis of 7 differential genes in UC and healthy controls (ggplo2 package mapping). ns $p>0.05$, $*p<0.05$, $**p<0.01$, $***p<0.001$, $****p<0.0001$. **(C)** 7 DEG-encoded protein interaction networks. The network nodes represent proteins, while the lines indicate predicted relationships: with light blue representing auxiliary database evidence, purple representing laboratory proof, yellow representing text mining evidence, green representing gene similarity, red representing gene fusion, blue representing gene co-production, black lines representing gene co-expression, and gray lines representing protein homology.

negatively correlated with the expression of Type 17 helper cells and CD56dim natural killer cells. Meanwhile, GXYLT2 and SEMA5A were significantly negatively correlated with the

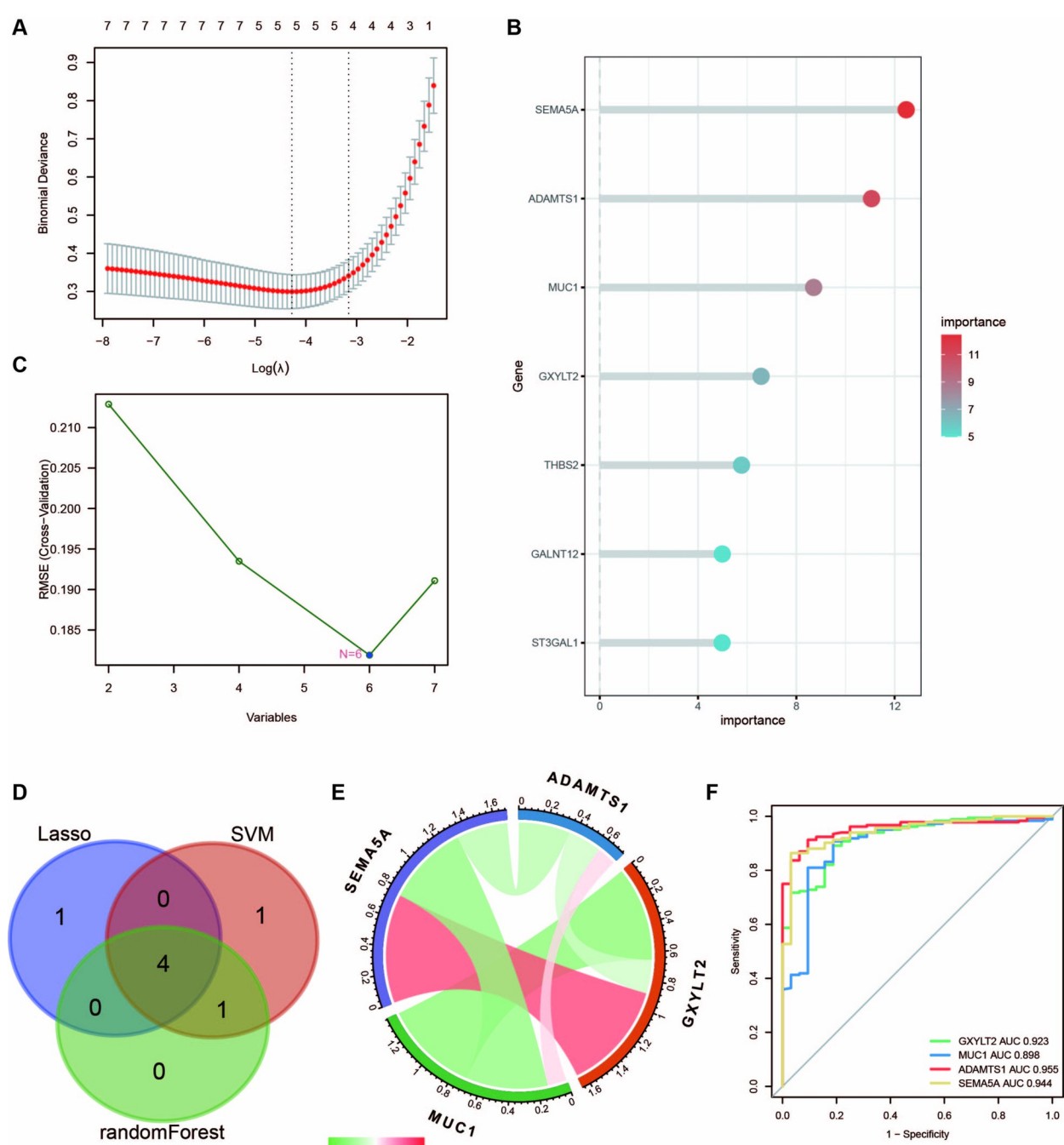

**Fig 4. Machine learning screening for key differential genes.** (A) LASSO regression screening of 5 genes. (B) RF selected 5 genes in order of importance. (C) SVM screened 6 genes. (D) Intersection obtained 4 core genes. (E) The correlation between the 4 core genes, with red represents a positive correlation, and green indicating a negative correlation. (F) ROC curve of 4 genes predicting disease occurrence.

expression of activated dendritic cells, activated CD4 T cells, natural killer cells, Type 17 helper cells, and CD56dim natural killer cells. Moreover, there was a significant positive correlation between the expression of MUC1 and activated dendritic cells, natural killer cells, activated CD4 T cells, and Type 17 helper cells (Fig 5C).

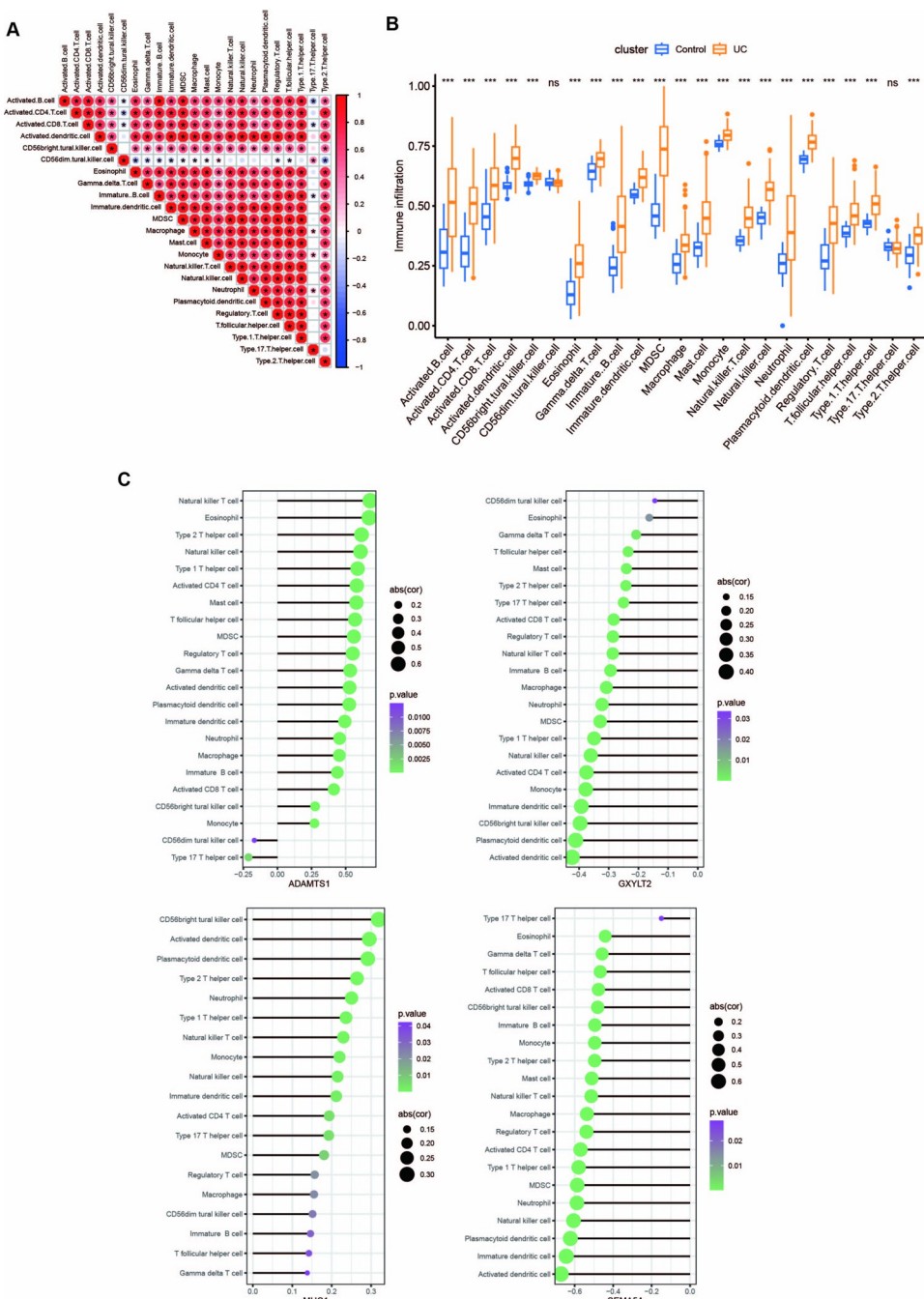

**Fig 5. Evaluation of the degree of immune cell infiltration. (A)** Correlation analysis between immune cells. **(B)** Differences in immune cell infiltration between UC and healthy control group, ns $p > 0.05$, *$p < 0.05$, **$p < 0.01$, ***$p < 0.001$. **(C)** Correlation analysis between 4 core genes and immune cells.

## Single gene enrichment analysis

Based on the significant role of the four hub genes, the correlation genes associated with ADAMTS1, GXYLT, MUC1, and SEMA5A expression were hereby analyzed. The heatmap positively revealed the top 50 co-expressed genes with four core genes (S2A–S2D Fig).

Single-gene GSEA was performed to characterize the potential function of the four hub genes. The ridgeline plot displayed only the top 20 results. Details are shown in Fig 6, and the values below representing enrichment scores, with a value exceeding 0 indicating a positive correlation between a gene and a pathway, while a value less than 0 indicating a negative correlation.

Almost all pathways identified were related to immunity and inflammation, including antigen processing-cross-presentation, signaling by interleukins, integral cell surface interactions, and interferon signaling. Meanwhile, MUC1 showed a negative correlation with both Asparagine N-linked glycosylation and O-linked glycosylation of mucins. Additionally, GXYLT2 and SEMA5A exhibited a negative correlation with collagen formation, whereas ADAMTS1 displayed a positive correlation with the same process (Fig 6A–6D).

## Unsupervised consensus clustering analysis of gene expression profiles revealed two subtypes of UC

An unsupervised consensus clustering analysis was conducted based on the four hub genes, with all UC samples initially divided into k (k = 2–9) clusters. The cumulative distribution function (CDF) curves of the consensus score matrix statistic indicating that the optimal number was obtained when k = 2. Consequently, two distinct subtypes of UC were identified (Fig 7A), involving 106 samples in subtype A and 78 in subtype B. The four genes exhibited remarkable differences in expression between the two subtypes (p<0.05). The expression of all other genes in subtype A was higher than that in subtype B, except for MUC1 (Fig 7B). Furthermore, a heatmap was drawn to more intuitively display the expression differences of four genes between 184 samples from two subtypes using the R software package "pheatmap". GXYLT2, ADAMTS1, and SEMA5A were significantly upregulated in subtype B, while MUC1 was upregulated considerably in subtype A, further validating the presence of diverse subtypes in UC (Fig 7C).

## GSVA of biological pathways between two subtypes

GSVA enrichment was performed to explore the biological behavior and pathway differences of the two clusters. The GSVA enrichment analysis showed that the two subtypes significantly varied in the metabolism of various substances. A heatmap of the genes was organized in ascending order according to their P values, and the top 20 were selected for further analysis.

The results of the KEGG analysis showed that the A subtype was enriched in pathways of base excision repair and substance metabolism, including galactose, fructose, mannose, amino sugar, nucleotide sugar metabolism, and glycerolipid. In contrast, the B subtype was frequently involved in cancer-related pathways, such as non-small cell lung cancer, colorectal cancer, chronic myeloid leukemia, endometrial cancer, among others. As a result, it could be reasonably speculated that subtype B of UC could possibly develop into ulcerative colitis-associated colorectal cancer (CAC) (Fig 8A).

Furthermore, the results of the Rectome analysis indicated that the subtype A was enriched in nucleotide catabolism and purine catabolism pathways. In contrast, the subtype B was enriched in pathways of ESR mediated signaling, signaling by the nuclear receptor, IGF1R signaling, RUNX2 regulates osteoblast differentiation, RUNX2 regulates bone development, and glutamate and glutamine metabolism (Fig 8B).

## Differential genes and enrichment analysis of the two subtypes

The principal component analysis (PCA) demonstrated that UC patients were well distributed into two clusters (Fig 9A). The PCA offered a holistic and clear visual representation, mapping

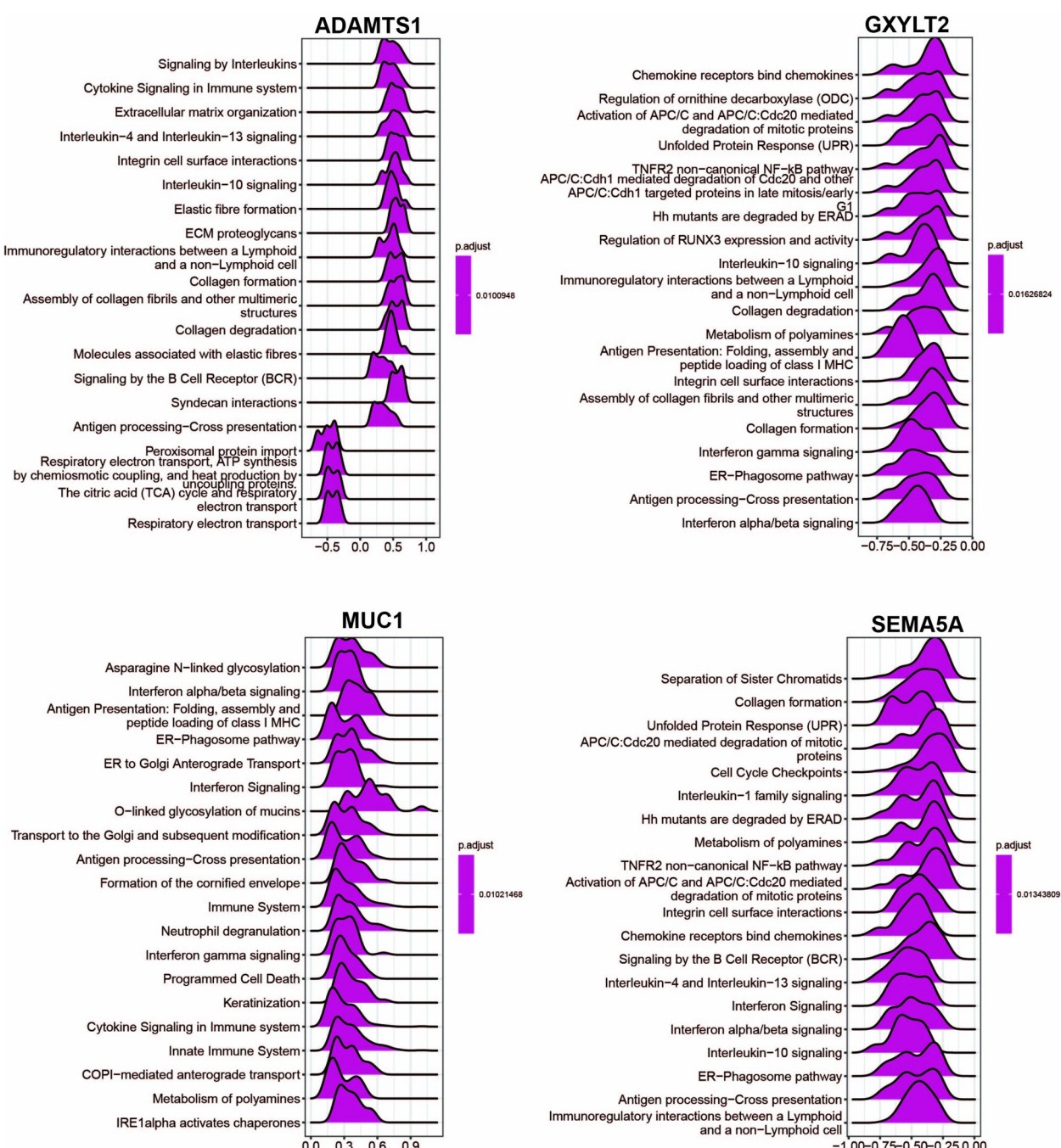

**Fig 6. Single gene enrichment analysis.** (**A**) GSEA analysis for ADAMTS1. (**B**) GSEA analysis for GXYLT2. (**C**) GSEA analysis for MUC1. (**D**) GSEA analysis for SEMA5A.

all samples and highlighting the separation between groups. The substantial distance between subtypes A and B indicated pronounced distinctions between them.

Through differentially expressed genes analysis, 229 DEGs were obtained, including 105 DEGs markedly upregulated and 124 DEGs significantly downregulated (Fig 9B). Following that, GO and KEGG analyses of DEGs were performed to further interpret the clustering results from the perspective of fundamental biological processes. The top ten results of GO enrichment

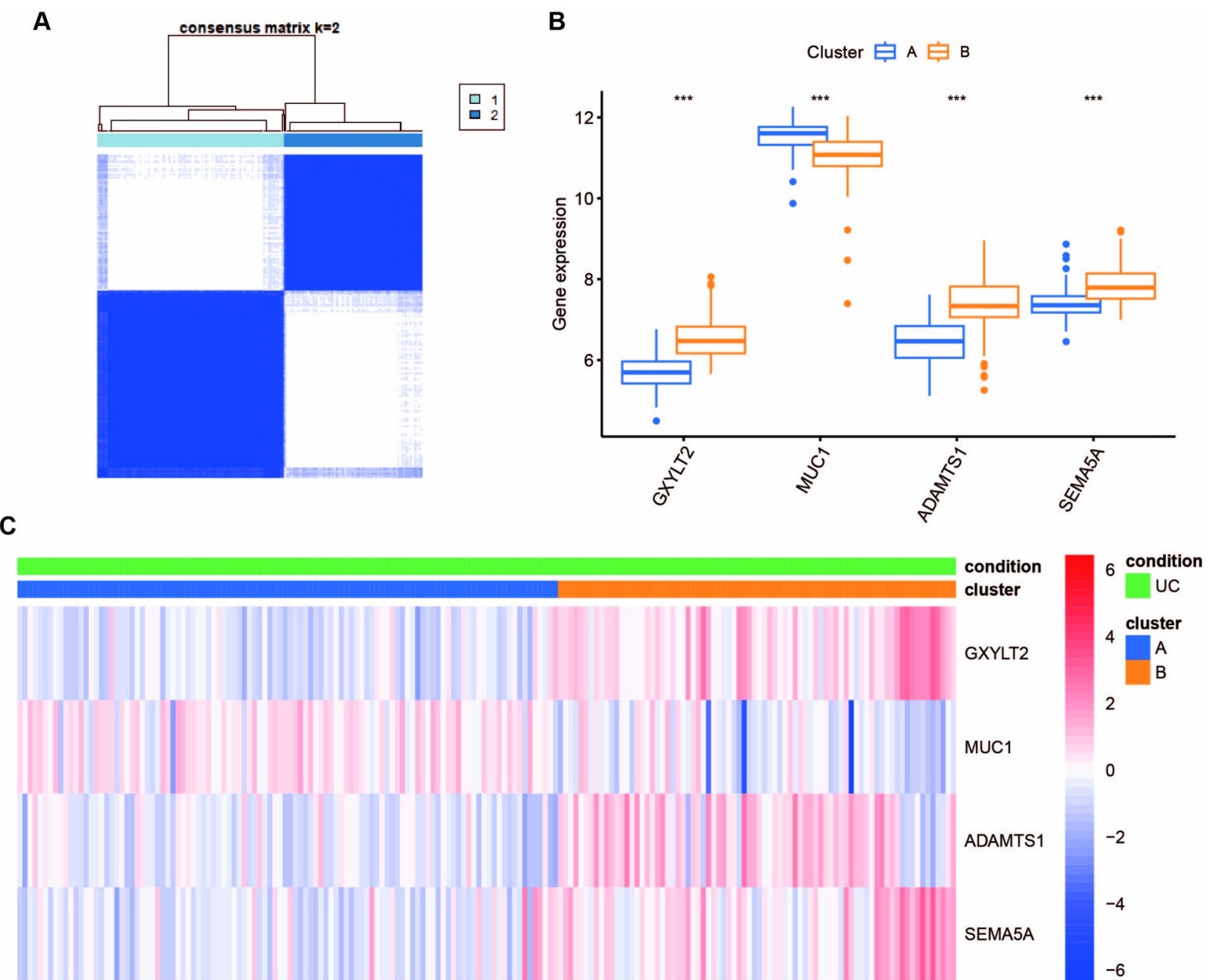

**Fig 7. Identification and validation of ulcerative colitis subtypes.** (**A**) Heatmap of sample clustering at consensus k = 2. (**B**) The expression status of four hub genes in the two subtypes, ***p<0.001. (**C**) Heatmap of four hub genes in the two subtypes.

analyses were exhibited, including BP, CC, and MF (Fig 9C). The BP indicated the enrichment function of the regulation of peptidase activity and response to peptide hormone. Meanwhile, the CC showed that the DEGs were primarily correlated with the collagen-containing extracellular matrix, apical part of the cell, and apical plasma membrane. For MF, extracellular matrix structural constituent, receptor ligand activity, and signaling receptor activator activity were mainly enriched for the DEGs. Additionally, KEGG analysis showed that the DEGs were primarily involved in inflammation, immunity, and infectious diseases (Fig 9D).

According to KEGG enrichment analysis, the top 5 significant pathways of DEGs and related genes were identified, including Cytokine-cytokine receptor interaction, IL-17 signaling pathway, Viral protein interaction with cytokine and cytokine receptor, Amoebiasis and Pertussis (Fig 9E) (Table 3).

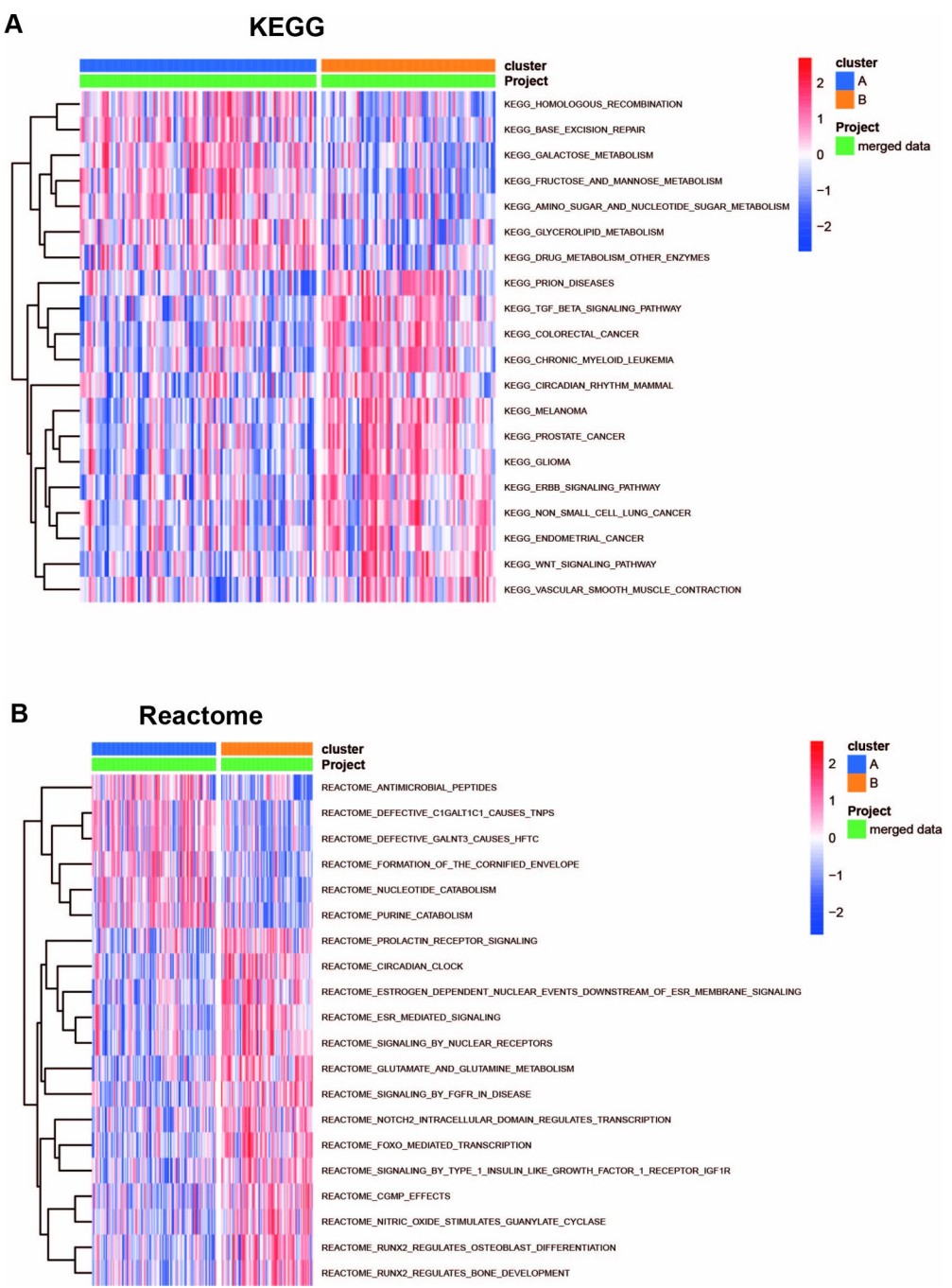

**Fig 8. The diversity of the underlying biological function characteristics between the two subtypes. (A)** The differences in KEGG pathway enrichment score between subtypes A and B. **(B)** The differences in Reactome pathway enrichment score between subtypes A and B.

## Prediction of miRNAs and transcription factors

To determine the upstream TFs and miRNAs of hub genes, 56 TFs and 49 miRNAs were obtained via the RegNetwork repository (https://regnetworkweb.org/), with a vast network established to present enhanced co-regulatory patterns using Cytoscape (Fig 10). MUC1, in the core position of the network, was regulated by 33 TFs and 27 miRNAs. For SEMA5A and

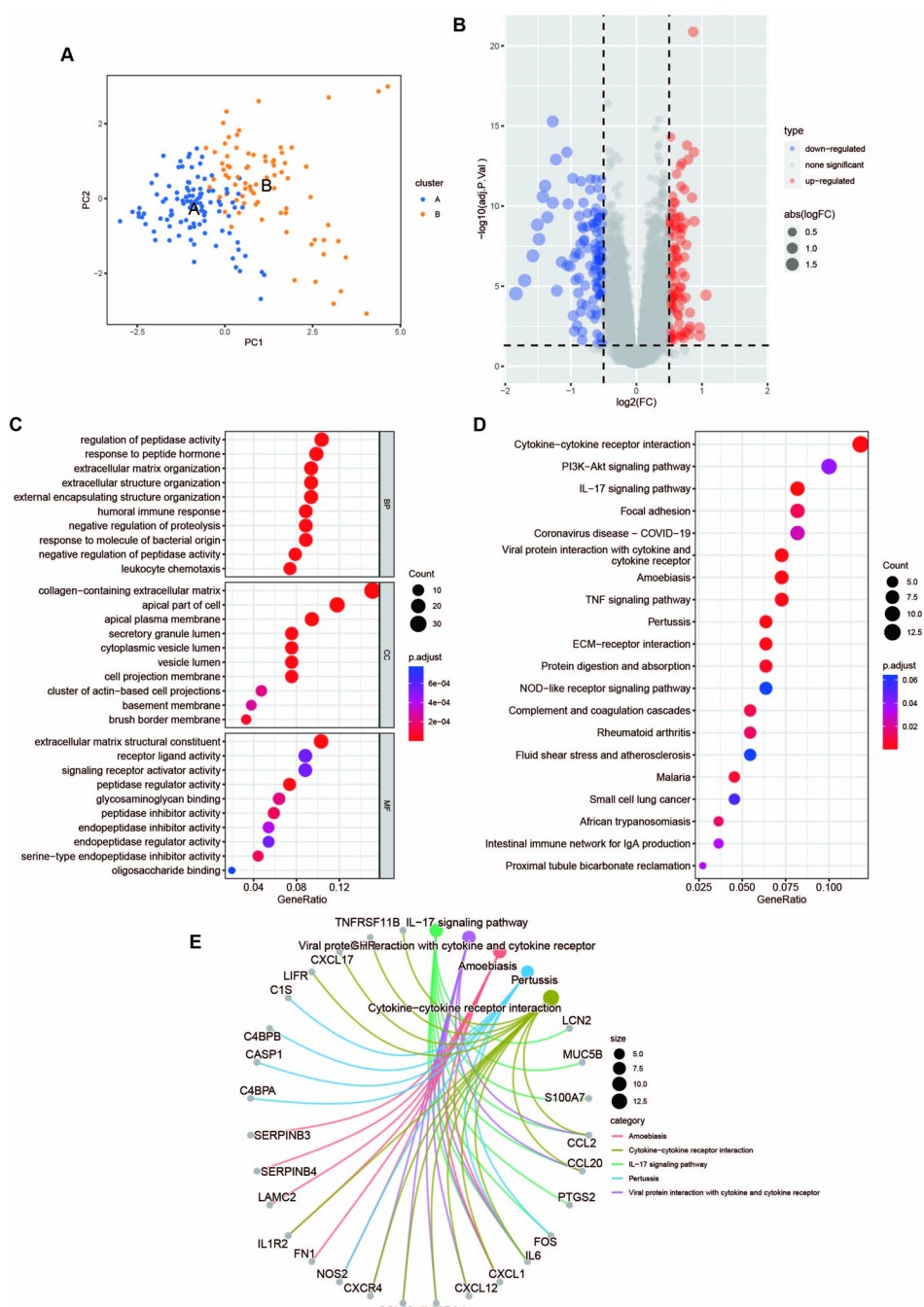

**Fig 9. Differential genes and enrichment analysis of the two subtypes.** (**A**) PCA analysis demonstrating a distinctive difference between the two clusters. (**B**) Volcano plot of the 229 DEGs. The threshold for the volcano plot was |logFC| >0.5 and adj.p.Val. < 0.05. (**C**) GO enrichment analysis showing the BP, CC, and MF parts. (**D**) The bubble plot depicting the KEGG pathway enrichment analysis of DEGs. (**E**) The correspondence between the KEGG top five pathways and genes.

MUC1, the common TF was SP1 and MEF2A, and miRNA was hsa-miR-519e. MUC1 and ADAMTS1 had four common TF, including TFAP2A, STAT1, STAT3, and CTCF. GXYLT2 had only one upstream miRNA has-miR-37 and two TF HNF4A and NR2F1.

## Discussion

Glycosylation is the process of attaching various sugars to proteins through glycosidic bonds, representing the most prevalent post-translational modification across all cellular organisms. Glycosylation enhances the stability of proteins, primarily involving N-glycans and O-glycans, with enzymes overseeing the entire process [10]. O-GlcNAc transferase (OGT) adds the O-linked β-N-acetylglucosamine (O-GlcNAc) monosaccharides to the serine or threonine residues of nuclear or cytoplasmic protein [35]. O-GlcNAcase (OGA) can then remove the monosaccharide reversibly [36]. Most protein glycosylation occurs in the endoplasmic reticulum (ER) and Golgi. O-glycosylation, in particular, regulates immune cells' development, homeostasis, and functions [37, 38].

As one type of IBD, UC is characterized by an abnormal immune response to the gut microbiota. The prevalence of UC is escalating year by year [39]. Mucosal lesions usually originate in the rectum and may spread to the entire colon as the disease progresses [3]. The extra-intestinal manifestations also influence the quality of life and even cause disability, including anemia, arthropathy, metabolic bone disease, and hepatobiliary disease [40]. Immune cells play an essential role in the occurrence and development of UC. Antigen-presenting cells (APCs), such as macrophages and dendritic cells (DCs), can recognize antigens and initiate the immune response by releasing cytokines like IL-12. L-12 is instrumental in driving the differentiation of Th1 cells, which in turn secrete the pro-inflammatory cytokines TNF-α, IFN-γ, and IL-2 [41, 42].

Gut microbiota influences intestinal physiology and emphasizes the potential of bacterial OGAs as a promising therapeutic strategy in colonic inflammation by hydrolyzing O-GlcNA-cylated proteins [43]. Moreover, Qian-Hui Sun et al. identified increased O-GlcNAc level in the gut epithelium of AIEC LF82-infected mice and CD patients, linking the change to intestinal inflammation [17]. In addition, in dextran sodium sulfate (DSS)-induced colitis and azoxymethane (AOM)/DSS-induced CAC mice models, the O-GlcNAcylation of colonic tissues was also elevated. Compared to normal colonic tissues, human CAC tissues' O-GlcNAcylation was increased [44]. Many studies have implicated O-GlcNAcylation as a contributing factor in the promotion of chronic colonic inflammation.

However, the relationship between O-GlcNAcylation and UC has not been well-studied, making it necessarily important to explore the specific molecular mechanism of O-GlcNAcylation in UC. Herein, efforts were made to determine the possible role of O-glycosylation in UC through bioinformatic analysis. Specifically, GSE75214 and GSE9241 datasets downloaded

**Table 3. The KEGG enrichment analysis table of the DEGs of the two subtypes.**

| ID | Description | GeneRatio | BgRatio | pvalue | p.adjust | geneID |
|---|---|---|---|---|---|---|
| hsa04657 | IL-17 signaling pathway | 9/110 | 94/9180 | 1.67E-06 | 0.000330056 | LCN2/MUC5B/S100A7/CCL2/CCL20/PTGS2/FOS/IL6/CXCL1 |
| hsa04061 | Viral protein interaction with cytokine and cytokine receptor | 8/110 | 100/9180 | 2.45E-05 | 0.001604571 | CXCL12/IL22RA1/CCL28/CCL2/CXCR4/CCL20/IL6/CXCL1 |
| hsa05146 | Amoebiasis | 8/110 | 102/9180 | 2.84E-05 | 0.001604571 | NOS2/FN1/IL1R2/LAMC2/SERPINB4/IL6/CXCL1/SERPINB3 |
| hsa05133 | Pertussis | 7/110 | 76/9180 | 3.24E-05 | 0.001604571 | C4BPA/NOS2/CASP1/C4BPB/C1S/FOS/IL6 |
| hsa04060 | Cytokine-cytokine receptor interaction | 13/110 | 295/9180 | 4.84E-05 | 0.00183806 | CXCL12/LIFR/IL1R2/CXCL17/IL22RA1/CCL28/GHR/CCL2/CXCR4/CCL20/TNFRSF11B/IL6/CXCL1 |

Table 3 demonstrated that KEGG top 5 pathways ranked in ascending order of p-value, and the relevant information including ID, Description, GeneRatio, BgRatio, pvalue, p.adjust, qvalue, geneID, and Count.

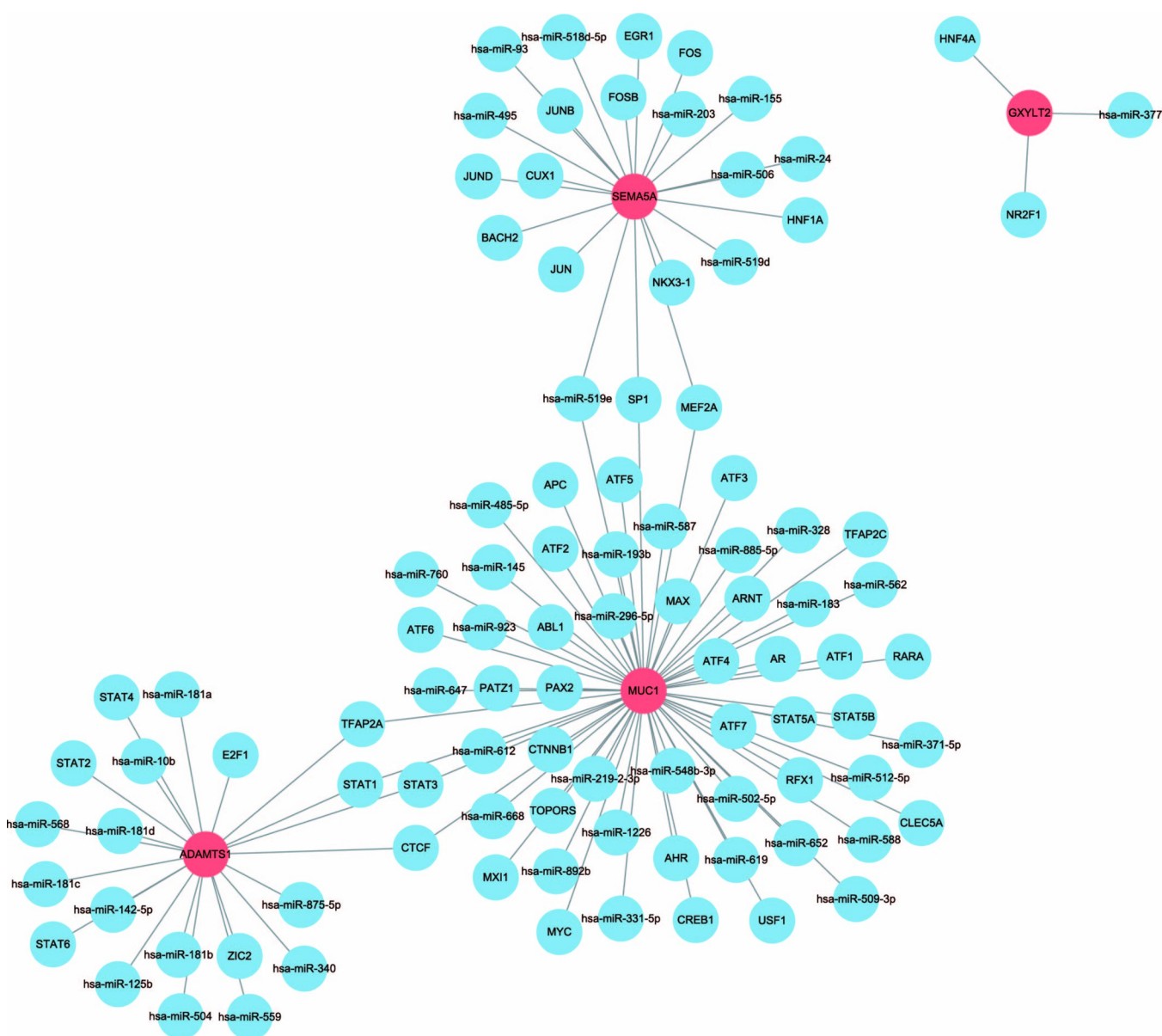

**Fig 10. TF–miRNA co-regulatory network analysis, with red nodes representing hub genes, and blue nodes indicating TFs and miRNAs.**

from GEO were analyzed to identify DEGs in UC patients. The GO and KEGG analyses revealed that the DEGs were enriched in immunity, inflammation, and cytokine signaling pathways.

To explore the relationship between O-GlcNAcylation and UC, the DEGs in UC were intersected with 151 O-GlcNAcylation-related genes. A total of 7 DEGs were detected. Furthermore, GO and KEGG were conducted for the seven DEGs. Through LASSO, SVM-RFE, and RF algorithm, four O-glycosylation-related hub DEGs in UC were screened, including MUC1, ADAMTS1, GXYLT2, and SEMA5A. The AUC values of the four hub genes were high, demonstrating these four hub genes as potential target genes for treating UC through O-glycosylation. This offered a new direction for exploring the role of O-glycosylation in UC. Given the importance of immunity in UC, an immune infiltration analysis was further performed. The

results revealed a significant difference between the normal group and UC patients. UC patients had a higher level of DCs, Th1 cells, Treg, B cell, CD4+ T cell, CD8+ T cell, macrophage and neutrophil compared to their normal counterparts. The results were highly consistent with the results of previous studies, underscoring the importance of immune cells in the pathogenesis of UC.

MUC1, a member of the mucin family and a membrane-bound protein, is secreted by goblet and absorptive cells of the intestinal epithelium and plays a role in the mucus layer [45]. It is highly expressed in the epithelial mucosa of the gastrointestinal tract. Mucins are O-glycosylated proteins that can form protective mucous barriers [46]. The membrane shift and overexpression of MUC1 affect the prognosis of related malignant tumors, like colon cancer [47, 48]. Besides, MUC1 also holds considerable significance in intracellular signaling and immune regulation, especially colonic inflammation [49–51]. Increased MUC1 expression is often associated with a decrease in the beneficial gut microbiota [52]. The breakdown of the mucus barrier and dysregulation of intestinal microflora can exacerbate the incidence and progression of UC. Yet, the precise mechanism of MUC1's involvement in UC requires additional research.

ADAMTS1, a disintegrin-like, and metalloprotease with the thrombospondin type 1 motif, is a protein-coding gene whose related pathways are the diseases associated with O-glycosylation. It plays a vital role in inflammatory processes and the development of cancer [53, 54] and presents angiogenic inhibitor activity [55]. Compared to the standard group, the ADAMTS1 level in UC patients was hereby found to be higher and correlated with IL-17. IL-17 may damage the intestinal wall by promoting the expression of ADAMTS1 [56]. However, the mechanisms by which ADAMTS1 operates in UC are not fully understood and warrant additional research for clarification.

GXYLT2 (glucoside xylosyltransferase) encodes a xylosyltransferase, which catalyzes the addition of xylose to the O-glucose-modified residues of EGF repeats of Notch proteins [57]. Compared to quiescent UC, the expression of GXYLT2 in active UC is elevated, which facilitates the assessment of disease activity [58]. Meanwhile, Barnicle et al. found that methylated genes GXYLT2 differed between inflamed tissues and regular counterparts of UC patients, and the Wnt signaling pathway was involved [59]. The dysregulation of Wnt and Notch signaling pathways, associated with the proliferation and differentiation of intestinal stem cells (ISCs), induces cell overgrowth and malignant transformation. In UC, the inhibition of Wnt and overexpression of Notch induce the decrease of Paneth cells, thereby leading to intestinal barrier damage [60].

The final gene among the four central hub genes is SEMA5A, with a limited number of studies currently available on this gene. Using differentially expressed lncRNAs to predict target genes, Benhai Xiong et al. investigated the effects of extracellular vesicles (EVs) on the expression of Sema5a genes in DSS-treated mice. The results showed a considerable down-regulation of Sema5a gene expression [61]. Besides, axon guidance cue Sema5a may cause the expression of pro-inflammatory genes (TNF-α and IL-8) [62].

Currently, UC classification is primarily based on the severity of the disease, categorized as mild, moderate, and severe. Classification at the genetic level, however, remains understudied. Therefore, UC was hereby further grouped into two subtypes using unsupervised consensus clustering. This classification was based on the expression of the four hub genes, utilizing machine learning methods and unsupervised clustering algorithms. The expression of the four hub genes varied significantly between the two subtypes. The four core gene expression trends in subtype B were the same as those of previous research. Furthermore, GSVA enrichment analysis showed that subtype A was enriched in various substance metabolisms, such as glucose metabolism, lipid metabolism, and amino acid metabolism. In contrast, subtype B was significantly enriched in cancer-related pathways, like colorectal cancer. This significant

finding suggested that subtype B of UC could potentially progress to CAC. Moreover, GO and KEGG enrichment analyses of subtype A and B DEGs were conducted to further identify the differences between the two subtypes. The results enriched in cytokine-related signaling pathways and immune-related diseases, underscore the significance of distinguishing between the two subtypes, calling for further research.

In conclusion, two subtypes of UC were hereby confirmed. Each possessed distinct molecular features, biological behavior, and clinical characteristics. Overall, the classification provides a basis for further studies about the therapy and prognosis of UC. However, this study is still subjected to several limitations. Firstly, a comparative analysis of survival curves for the two UC subtypes was not conducted. Secondly, the study relied solely on bioinformatic methods, which have yet to be experimentally validated. The sampling method did not exclude the effects of age, gender, disease severity, complications, and therapeutic approaches. Further efforts could be made to compare the hub gene expression level between the mice with DSS and the control group, UC patients, and healthy individuals, and to explore the prognosis of different subtypes of UC and their effects on CAC.

In summary, the research warrants in-depth exploration to demonstrate the further mechanisms of O-GlcNAcylation, the expression of the hub genes, and the clinical significance of the two subtypes in UC.

## Supporting information

**S1 Fig. The two databases underwent normalization and subsequent merging.** (A-B) GSE75214 and GSE92415 combined and used R packages "limma" and "sva" to remove batch effects, resulting in 16,467 genes and 216 samples. A: before merging; and B: after merging. (C-D) R language preprocessCore package homogenized the dataset; C: before homogenization; and D: after homogenization.
(TIF)

**S2 Fig. The correlation analysis between the four hub genes and all other genes.** The positive correlation between the top 50 genes and 4 hub genes was displayed using heatmaps.
(TIF)

**S1 File. DEGs of GSE75214 and GSE92415.** The genes from both the UC patients and healthy individuals in the GSE75214 dataset were merged with those from GSE92415 to form a comprehensive data set. The batch effect was then eliminated to minimize discrepancies between the different datasets. There were a total of 16467 genes after combination.
(CSV)

**S2 File. R code.** This is the R language code used by the bioinformatics method involved in this study.
(DOCX)

## Acknowledgments

The authors would like to thank Shuyuan Zhang of the First Affiliated Hospital of Harbin Medical University for helpful discussions on topics related to this work. Besides, I could not have completed this dissertation without the support of my friends Dr. Yonghui Wu, who provided stimulating discussions and creative ideas.

## Author Contributions

**Conceptualization:** Yue Lu, Hongyu Xu.

**Funding acquisition:** Dongyue Li.

**Investigation:** Dongyue Li.

**Methodology:** Yue Lu, Yi Su, Nan Wang.

**Visualization:** Yi Su, Nan Wang.

**Writing – original draft:** Yue Lu, Yi Su, Nan Wang, Huichao Zhang.

**Writing – review & editing:** Yue Lu.

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
