## [Decision Letter · Decision Letter 0]

9 May 2024

PONE-D-24-08842Identification of O-Glycosylation related genes and subtypes in Ulcerative Colitis based on machine learningPLOS ONE

Dear Dr. Xu,

Thank you for submitting your manuscript to PLOS ONE. After careful consideration, we feel that it has merit but does not fully meet PLOS ONE’s publication criteria as it currently stands. Therefore, we invite you to submit a revised version of the manuscript that addresses the points raised during the review process.

We look forward to receiving your revised manuscript.

Kind regards,

Ashutosh Pandey, Ph.D.

Academic Editor

PLOS ONE

Journal Requirements:

Reviewers' comments:

Reviewer's Responses to Questions

**Comments to the Author**

1. Is the manuscript technically sound, and do the data support the conclusions?

Reviewer #1: Yes

Reviewer #2: Partly

2. Has the statistical analysis been performed appropriately and rigorously? 

Reviewer #1: Yes

Reviewer #2: Yes

3. Have the authors made all data underlying the findings in their manuscript fully available?

Reviewer #1: Yes

Reviewer #2: Yes

4. Is the manuscript presented in an intelligible fashion and written in standard English?

Reviewer #1: No

Reviewer #2: No

5. Review Comments to the Author

Reviewer #1: In the present manuscript, the authors attempted to demonstrate the role of O-Glycosylation-related genes and subtypes like MUC1, ADAMTS1, GXYLT2, and SEMA5A in Ulcerative Colitis using bioinformatics approaches. The manuscript is good but required major revisions prior to the acceptance.

1. The quality of the writing is extremely poor and lacks scientific explanation. The choice of words for discussing any information requires more background studies and details technical descriptions. The abstract needed to be reframed and the introduction also required substantial corrections.

2. The background knowledge for the selection of the two different datasets for the study viz., GSE75214 and GSE92415 is not clear. The patient data from the selected population needed to be properly demonstrated. The authors are requested to follow https://doi.org/10.1371/journal.pone.0289064 for their reference. It was found that one of the datasets GSE92415 contained the data for UC patients treated with Golimumab, it is not mentioned anywhere in the manuscript and the rationale behind the selection of this particular study is not clear in the manuscript. It needed to be addressed properly.

3. The selection of DEGs from the two discrete datasets is not clear. The authors should either first select the common DEGs between both the datasets and then screen for the Upregulated and downregulated genes or clearly describe their criteria for the selection process. The statistical significance and relevance of the screening and functional enrichment needed to be justified properly. Similar conditions apply to the LASSO analysis.

4. The resolution of the figures needed to be modified along with the figure captions. The font and size of the text used in the figure are unclear and need to be modified.

5. Regarding the establishment of the role of the critical genes in CRC/CAC, the authors may explore the differential expression of these critical genes for human CRC using the TCGA data. The authors can follow https://doi.org/10.1016/j.humgen.2023.201189 and https://doi.org/10.3389/fgene.2021.608313 for their reference.

Reviewer #2: The study aims to investigate the role of altered intestinal glycosylation, particularly O-GlcNAcylation modification of proteins in the pathogenesis of Ulcerative colitis (UC) which could have potential implications for diagnosis and treatment. For this purpose, the study effectively integrates previously published transcriptional and clinical data by employing various machine learning methods. The article identifies MUC1, ADAMTS1, GXYLT2 and SEMA5A as significantly associated with UC-related O-GlcNAcylation. The authors propose two UC subtypes based on the expression of these four hub genes. Interestingly, subtype B (defined by elevated expression of ADAMTS1, GXYLT2 and SEMA5A) shows a potential predisposition to colitis-associated colorectal cancer (CAC), providing valuable insights into disease progression.

Strengths:

This study used a systemic data-driven approach to address an important question of the role of O-GlcNAcylation in UC pathogenesis and progression, offering a foundation for further research. The identification of two UC subtypes based on the four hub genes represents a significant contribution to the field.

Concerns:

I have a few concerns and suggestions regarding this article in its present form, most which stem from the presentation of the results (such as by adding key details of the bioinformatic analysis to the text). Addressing these concerns will enhance the article’s clarity and enable the readers to better assess the study’s findings.

Major Issues:

1. Gene lists should be provided for differential expression analysis along with logFC values and p-values (e.g. for Figure 1 and 2).

2. For all figures, make sure to specify a legend for the scale used and any relevant cutoffs for fold change or p-value in the figure description (e.g for the heatmap in Figure 1). Furthermore, make sure to provide which figure you are referring to in the text (line 418).

3. There is some ambiguity in the description of analysis (e.g lines lines 427 to 429). For figure 4A-C, specify which tag genes were identified by each of the algorithms in the figures. The number of genes identified is inconsistent with the in-text description of the figure (lines 128 to 131). Furthermore, Figure 4E could be described in more detail in the text (specify positive/negative correlation or the exact correlation value).

4. Relevant citations should be provided throughout the paper for all claims, (e.g. lines 25-30 and 43-45). Also provide references for packages and databases used (e.g. lines 152 and 427).

5. The discussion of the results in the context of prior literature on UC would be of immediate interest for biologists and clinicians. In this regard, further development of the implications of the results can strengthen the paper.

Minor Issues:

1. The abstract and introduction mention the etiology of UC and the role of O-GlcNAcylation in various inflammatory diseases. However, the context from prior literature that the authors already provide could be strengthened by providing further details about which immune cells are known to play a role in promoting mucosal immune and inflammatory responses in UC.

2. In the introduction, the authors mention that the level of protein O-GlcNAc changes (lines 32-35). Discuss these alterations and their implication further.

3. In the results section, the article should introduce the two datasets before jumping into the analysis in the results section. Explicitly state what the differential expression analysis is comparing (e.g. UC vs. healthy controls in line 51) to better guide the reader.

4. Make sure the nomenclature for human genes/transcripts/proteins is correct throughout the manuscript text and figures (e.g. in gene lists provided adjacent to heatmaps).

5. Discuss the limitations of the study and suggest future experiments that can validate the findings.

6. Providing a table of the pathways along with p-values for the pathway enrichment analysis would be helpful.

7. Some of the ambiguity in the analysis may be addressed by providing the code for replication by others.

8. Language can be improved for clarity (e.g. lines 23-25). Please also proofread for typos (e.g. line 26 and 29).

6. PLOS authors have the option to publish the peer review history of their article (what does this mean?). If published, this will include your full peer review and any attached files.

Reviewer #1: No

Reviewer #2: **Yes: **Anukriti Singh

---

## [Author Response · Author response to Decision Letter 0]

19 Jun 2024

Dear Editor and Reviewer:

Thank you for your decisions and constructive comments on our manuscript entitled “Identification of O-Glycosylation related genes and subtypes in Ulcerative Colitis based on machine learning”. Those comments are all valuable and very helpful for revising and improving our paper. We have studied comments carefully and have made corrections which we hope meet with approval. The reviewers’ comments are laid out below in italicized font and specific concerns have been numbered. Our response is given in normal font and changes/additions to the manuscript are given in the red text.

Responses to Reviewers comments:

Reviewer #1:

1. The quality of the writing is extremely poor and lacks scientific explanation. The choice of words for discussing any information requires more background studies and details technical descriptions. The abstract needed to be reframed and the introduction also required substantial corrections.

We apologize for the poor language of our manuscript. We have now worked on both language and readability, and really hope that the flow and language level have been substantially improved.

2.The background knowledge for the selection of the two different datasets for the study viz., GSE75214 and GSE92415 is not clear. The patient data from the selected population needed to be properly demonstrated. The authors are requested to follow https://doi.org/10.1371/journal.pone.0289064 for their reference. It was found that one of the datasets GSE92415 contained the data for UC patients treated with Golimumab, it is not mentioned anywhere in the manuscript and the rationale behind the selection of this particular study is not clear in the manuscript. It needed to be addressed properly.

We are sorry for our carelessness. In the methods section, we provide detailed background on the databases GSE75214 and GSE92415. In addition, the data for UC patients treated with Golimumab were not included in our study (Line 77-87).

This study incorporated two datasets, GSE75214 (provided by the GPL6244 platform) and GSE92415 (provided by the GPL13158 platform). The GSE75214 database contains intestinal mucosal biopsies obtained endoscopically from UC patients (n=97) and healthy controls (n=11), and subsequently analyzed for gene expression by microarray. GSE92415 database enrolled 21 healthy subjects and 162 UC patients, including baseline before treatment (n=87) and post-treatment individuals (n=75), to evaluate the effect of golimumab (GLM) during induction treatment in moderately to severe UC. In this study, we selected only 87 untreated UC samples and 21 healthy control samples.

3.The selection of DEGs from the two discrete datasets is not clear. The authors should either first select the common DEGs between both the datasets and then screen for the Upregulated and downregulated genes or clearly describe their criteria for the selection process. The statistical significance and relevance of the screening and functional enrichment needed to be justified properly. Similar conditions apply to the LASSO analysis.

We apologize for the lack of clear instructions on the differential genes screening process. In order to increase the sample size, we included multiple datasets. However, there were batch effects among these datasets due to various factors, such as experimental time, batch, laboratory, and sample processing method. These batch effects were combined and eliminated to minimize technical differences and ensure data reliability and comparability. We combined GSE75214 and GSE92415 datasets and performed differential analysis using the R language limma package. According to the criteria of |logFC|>1 and adj.P. Val<0.05, 449 genes were up-regulated, and 233 genes were down-regulated.

4.The resolution of the figures needed to be modified along with the figure captions. The font and size of the text used in the figure are unclear and need to be modified.

We modified the captions of several figures to make them more accurate, like Figure 4, and the resolutions are also adjusted appropriately. If any figure is not up to standard, please point it out and we will carefully modify it until it meets the PLOS ONE's standards. We sincerely thank the editor and all reviewers for the valuable feedback.

5. Regarding the establishment of the role of the critical genes in CRC/CAC, the authors may explore the differential expression of these critical genes for human CRC using the TCGA data. The authors can follow https://doi.org/10.1016/j.humgen.2023.201189 and https://doi.org/10.3389/fgene.2021.608313 for their reference.

We compared the expression of four hub genes in the TCGA and GEPIA databases, and found the differential expression is not exactly consistent with our study, as shown in the following pictures. For CRC studies, the expression levels of four key genes were also different in the two databases. In the TCGA database, the expression levels of four key genes in CRC were lower than those in the control group. Whereas, in the GEPIA database, MUC1, GXYLT2, and SEMA5A were highly expressed in CRC, thereinto SEMA5A had an obvious difference. ADAMTS1 was expressed at a low level in colorectal cancer. However, it is worth noting that the four core gene expression trends in subtype B were the same as those in previous research on UC, which we discussed amply in the Discussion section. 

Colorectal cancer (CRC) and colitis-associated colorectal cancer (CAC) have crucial differences. CAC occurs due to complications of chronic inflammatory disorders of the colon, while most CRC arise from precancerous lesions or adenomatous polyps[1]. However, there are only CRC and no CAC data in TCGA, so we did not show the differential expression of these critical genes in the manuscript. 

If the reviewers have better suggestions, we hope to strive for another opportunity to make modifications. 

[1] Kasi A, Handa S, Bhatti S, Umar S, Bansal A, Sun W. Molecular Pathogenesis and Classification of Colorectal Carcinoma. Curr Colorectal Cancer Rep. 2020 Sep;16(5):97-106. doi: 10.1007/s11888-020-00458-z. Epub 2020 Aug 15. PMID: 32905465; PMCID: PMC7469945

Reviewer #2：

Thanks for your thoughtful review. In the results section, we do have a lot of shortcomings. We have made detailed revisions and added the details of bioinformatic analysis to make the logic more fluent and clearer.

1. Gene lists should be provided for differential expression analysis along with logFC values and p-values (e.g. for Figure 1 and 2).

Because the gene lists are too large to be displayed directly in the manuscript, they are uploaded as supplementary tables along with logFC values and p-values. In addition , we added the KEGG enrichment analysis tables of DEGs in the manuscript, as shown in Table 1 and 2.

2. For all figures, make sure to specify a legend for the scale used and any relevant cutoffs for fold change or p-value in the figure description (e.g for the heatmap in Figure 1). Furthermore, make sure to provide which figure you are referring to in the text (line 418).

We apologize for lack of rigor. In the revised manuscript, we have added the logFC values and p-values, mainly Figure 1 and 9. Regarding the original 418 line section, we have reorganized the language and added the figure to make the description clearer (Line 97-100).

3. There is some ambiguity in the description of analysis (e.g. lines 427 to 429). For figure 4A-C, specify which tag genes were identified by each of the algorithms in the figures. The number of genes identified is inconsistent with the in-text description of the figure (lines 128 to 131). Furthermore, Figure 4E could be described in more detail in the text (specify positive/negative correlation or the exact correlation value).

In the section on methods and results of machine learning, we have provided a more detailed explanation (Line 128-135). For Figure 4A-C, the genes identified by each algorithm are listed in the revised manuscript. The explanation of Figures 4E and F has also been added (Line 307-319).

4. Relevant citations should be provided throughout the paper for all claims, (e.g. lines 25-30 and 43-45). Also provide references for packages and databases used (e.g. lines 152 and 427).

Based on your suggestions, we have checked the literature carefully and added more references on the definition of Glycosylation and the relationship between O-Glycosylation and diseases into the introduction part of the revised manuscript (Line 40-64). In Materials and methods part, we hope the complements of R packages references and websites of the databases involved can improve the credibility of the paper and meet the requirements of your journal. Besides, the key R packages have been uploaded as supporting information named “R packages”.

5. The discussion of the results in the context of prior literature on UC would be of immediate interest for biologists and clinicians. In this regard, further development of the implications of the results can strengthen the paper.

At the beginning of the discussion, background information on ulcerative colitis was added (Line 510-521). The intestinal symptoms and complications of ulcerative colitis can reduce the quality of life of patients, so it is necessary to explore its pathogenesis to guide treatment. We hope that the text added can arouse the interest of readers and look forward to further guidance from the reviewers.

Minor Issues:

1. The abstract and introduction mention the etiology of UC and the role of O-GlcNAcylation in various inflammatory diseases. However, the context from prior literature that the authors already provide could be strengthened by providing further details about which immune cells are known to play a role in promoting mucosal immune and inflammatory responses in UC.

According to the suggestions of the two reviewers, the abstract and introduction of this paper have been greatly improved. In the introduction and discussion sections, the correlation between immune cells and UC was supplemented (Line 31-35).

2. In the introduction, the authors mention that the level of protein O-GlcNAc changes (lines 32-35). Discuss these alterations and their implication further.

Thank you for the comment, the alterations and implications of O-GlcNAc changes are further described in lines 47-52.

3. In the results section, the article should introduce the two datasets before jumping into the analysis in the results section. Explicitly state what the differential expression analysis is comparing (e.g. UC vs. healthy controls in line 51) to better guide the reader.

We apologize for not providing a detailed introduction to the two datasets. Due to adjusting the order of methods and results, the background information of the datasets is presented in the methods section (Line 77-87), including methods and R packages for merging two datasets and screening for differentially expressed genes (Line 89-100).

4. Make sure the nomenclature for human genes/transcripts/proteins is correct throughout the manuscript text and figures (e.g. in gene lists provided adjacent to heatmaps).

 Thanks for your reminder, the nomenclature of human genes/transcripts/proteins in the text and figures are correct after checking.

5. Discuss the limitations of the study and suggest future experiments that can validate the findings.

 In the original manuscript, the limitations were indeed a little brief. Now we have further discussed the limitations and added what experimental methods can be further supplemented in the future (Line 626-635).

6. Providing a table of the pathways along with p-values for the pathway enrichment analysis would be helpful.

The tables of the pathways along with p-values for the pathway enrichment analysis have been uploaded. 

7. Some of the ambiguity in the analysis may be addressed by providing the code for replication by others.

 We have provided the code by which the steps of this article can be repeated.

8. Language can be improved for clarity (e.g. lines 23-25). Please also proofread for typos (e.g. line 26 and 29).

We sincerely thank the reviewer for careful reading. In the resubmitted manuscript, the typo is revised, and the grammar is corrected.

We tried our best to improve the manuscript and made some changes, shown in detail in the file “Revised Manuscript with Track Changes”. In addition, we would like to further confirm the authors’ order of the paper “Identification of O-Glycosylation related genes and subtypes in Ulcerative Colitis based on machine learning”. The following three authors, Yue Lu, Yi Su, and Nan Wang, contributed equally to this work and be considered co-first authors. In addition, Dongyue Li was added as the 2nd set of equal contributors, whose e-mail is “lidongyue83@163.com”, and the author Shuyuan Zhang was deleted. We appreciate for editors and reviewers’ work earnestly, and hope correction will meet with approval. Once again, thank you very much for your comments and suggestions.

I appreciate your consideration. I am looking forward to hearing from you. 

Sincerely, 

Hongyu Xu 

Institution and address: the First Affiliated Hospital of Harbin Medical University, 23 You Zheng Street, Nangang District, Harbin City, Heilongjiang Province, China

Telephone: 86-13903656899 

E-mail: xuhongyu@ldy.edu.rs

---

## [Decision Letter · Decision Letter 1]

18 Jul 2024

PONE-D-24-08842R1Identification of O-Glycosylation related genes and subtypes in Ulcerative Colitis based on machine learningPLOS ONE

Dear Dr. Xu,

Thank you for submitting your manuscript to PLOS ONE. After careful consideration, we feel that the revised version is improved but needs some revision to fully meet PLOS ONE’s publication criteria. Therefore, we invite you to submit a revised version of the manuscript that addresses the points raised during the review process.

We look forward to receiving your revised manuscript.

Kind regards,

Ashutosh Pandey, Ph.D.

Academic Editor

PLOS ONE

Journal Requirements:

Reviewers' comments:

Reviewer's Responses to Questions

**Comments to the Author**

1. If the authors have adequately addressed your comments raised in a previous round of review and you feel that this manuscript is now acceptable for publication, you may indicate that here to bypass the “Comments to the Author” section, enter your conflict of interest statement in the “Confidential to Editor” section, and submit your "Accept" recommendation.

Reviewer #2: (No Response)

2. Is the manuscript technically sound, and do the data support the conclusions?

Reviewer #2: Yes

3. Has the statistical analysis been performed appropriately and rigorously? 

Reviewer #2: Yes

4. Have the authors made all data underlying the findings in their manuscript fully available?

Reviewer #2: Yes

5. Is the manuscript presented in an intelligible fashion and written in standard English?

Reviewer #2: No

6. Review Comments to the Author

Reviewer #2: There are still come minor lapses in writing such as poor word choice and grammar, which should be fixed prior to publication. For instance, typos like "For visualization, the DEGs," (line 104) and numerous others should be corrected. Phrases like “involving three low expressions.” should be fixed to be informative and grammatically correct. Also the font changes give a sloppy appearance in "Identification of differential expressed genes".

Tables for GO analysis are not legible (exclude extraneous information) or shown as bar charts, with tables presented as supplementary information. Discussion has been thoroughly improved and comments sufficiently addressed.

7. PLOS authors have the option to publish the peer review history of their article (what does this mean?). If published, this will include your full peer review and any attached files.

Reviewer #2: **Yes: **Anukriti Singh

---

## [Author Response · Author response to Decision Letter 1]

29 Aug 2024

Dear Editor and Reviewer:

We thank the reviewer for the kind consideration and constructive comments on our manuscript. We have addressed your concerned in a point-by-point manner below, and hope that you will find the added information suitable and sufficient for publication. The reviewers’ comments are laid out below in italicized font and specific concerns have been numbered. Our response is given in normal font.

1.Typos like "For visualization, the DEGs," (line 104) and numerous others should be corrected. 

Grammatically, our expression lacks rigor and we have made changes to make it more precise.(line104-105)

2.Phrases like “involving three low expressions.” should be fixed to be informative and grammatically correct. 

 Our representation is not accurate and concise enough. Considering Figure 3 has clearly shown the differential expression of genes between UC patients and controls, we deleted “involving three low expressions.”, which does not affect the reader's understanding.  

3.The font changes give a sloppy appearance in "Identification of differential expressed genes".

In paragraph "Identification of differential expressed genes", the font is unified as “Times New Roman”, and the size is 14.

4.Tables for GO analysis are not legible (exclude extraneous information) or shown as bar charts, with tables presented as supplementary information. 

According to the reviewer's suggestion, “qvalue” and “count” were deleted in the kegg tables to make the forms more concise.

We sincerely appreciate the time and effort invested by the reviewers in evaluating our manuscript. We are more than happy to make any further revisions to improve the paper and facilitate successful publication. Once again, thank you very much for your comments and suggestions.

Sincerely, 

Hongyu Xu 

Institution and address: the First Affiliated Hospital of Harbin Medical University, 23 You Zheng Street, Nangang District, Harbin City, Heilongjiang Province, China

Telephone: 86-13903656899 

E-mail: xuhongyu@ldy.edu.rs

---

## [Editor Report · Decision Letter 2]

18 Sep 2024

Identification of O-Glycosylation related genes and subtypes in Ulcerative Colitis based on machine learning

PONE-D-24-08842R2

Dear Dr. Xu,

We’re pleased to inform you that your manuscript has been judged scientifically suitable for publication and will be formally accepted for publication once it meets all outstanding technical requirements.

Kind regards,

Ashutosh Pandey, Ph.D.

Academic Editor

PLOS ONE
---

## [Editor Report · Acceptance letter]

6 Dec 2024

PONE-D-24-08842R2 

PLOS ONE

Dear Dr. Xu, 

I'm pleased to inform you that your manuscript has been deemed suitable for publication in PLOS ONE. Congratulations! Your manuscript is now being handed over to our production team.

Kind regards, 

on behalf of

Dr. Ashutosh Pandey 

Academic Editor

PLOS ONE